# Continuum of community engagement to ensure access to health care in conflict-affected areas in South Sudan and Central African Republic (CAR): Lessons from the Red Cross

**Akalewold T. Gebremeskel**[1,2,3]*, **Puspita Hossain**[1,4], **Ilja Ormel**[1], **Faiza Rab**[1], **Mekdes Assefa**[1], **Christina Angelakis**[1], **Mariam Kone**[1], **Salim Sohani**[1]

**1** Global Health and Research Department, Canadian Red Cross, Ontario, Canada, **2** School of Nursing, Faculty of Health Sciences, University of Ottawa, Ottawa, Ontario, Canada, **3** School of International Development and Global Studies, University of Ottawa, Ottawa, Ontario, Canada, **4** Health Research Methods, Evidence, and Impact, McMaster University, Hamilton, Ontario, Canada

* agebr013@uottawa.ca

## Abstract

In conflict-affected settings such as South Sudan and the Central African Republic (CAR), fragile health systems face immense challenges in maintaining service delivery including essential health service access. Community engagement is a critical enabler across the Disaster Risk Management (DRM) cycle—preparedness, response, recovery, and resilience—by fostering trust, inclusivity, and local ownership. This qualitative study draws on data from the Advanced Partnership in Health (APiH) program (2019–2023), implemented by the Red Cross in CAR and South Sudan. Data were collected through 15 key informant interviews and 16 focus group discussions with community members, health workers, and program implementers. Framework analysis was used to explore the role of community engagement across each DRM phase. Community engagement was integral across all DRM phases—preparedness, response, recovery, and resilience. It enabled trust-building, inclusive participation, support for community health services, and local resource mobilization, contributing to more effective and sustainable health interventions in conflict settings. Community engagement is essential for sustaining healthcare delivery in conflict settings. Humanitarian health programs should institutionalize community engagement throughout the DRM cycle. Policies must prioritize inclusive planning, support for community health workers, and investment in local infrastructure to build resilient, community-driven health systems.

## Introduction

Humanitarian crises, particularly complex emergencies, have significantly increased over the past few decades [1,2].In 2024, the world saw 61 state-based armed

**Data availability statement:** We have included most of the required data for this paper. Due to ethical considerations and the involvement of human research participants, we are unable to share the full dataset publicly. However, additional data supporting the findings of this study can be made available upon reasonable request through the Canadian Red Cross data repository. Please contact communications@redcross.ca or Faiza Rab Faiza.Rab@redcross.ca Phone: 1 3653234074.

**Funding:** The funding for this work was provided through the APiH project financially supported by CRC and ICRC.

**Competing interests:** The authors have declared that no competing interests exist.

conflicts across 36 countries—the highest number recorded since 1946. This surge reflects a structural shift in global violence, with overlapping conflicts and transnational actors making humanitarian response increasingly difficult. The past three years have also been the most violent in three decades, with annual battle-related deaths exceeding 120,000 [3]. Complex emergencies arise from situations such as authority breakdowns, attacks on critical locations, and various conflict scenarios [4]. The resulting disruption of social networks and community infrastructure in such settings hamper health service delivery [5,6]. For example, protracted armed conflicts in South Sudan and the Central African Republic (CAR) have severely weakened health systems and infrastructure. In these contexts, humanitarian interventions prove to be essential for maintaining access to healthcare services [1,7].

Community engagement is a pillar of humanitarian response, particularly in conflict-affected settings where fragile health systems face immense challenges to health and other social service access due to security threats, infrastructure destruction, service disruptions, and population displacement. It fosters a two-way dialogue between humanitarian organizations and crisis-affected communities, ensuring that local needs, vulnerabilities, and capacities are effectively understood and addressed [8]. Additionally, it empowers communities by enhancing access to critical information, creating opportunities for meaningful feedback, and strengthening mutual accountability between stakeholders [9].

The barriers to community engagement include logistical constraints, cultural and linguistic diversity, restricted access to conflict zones for the service providers, and mistrust between communities and external actors [10]. Engaging local communities can provide cultural and contextual insights as well as critical insights into the unique needs before, during, and after a conflict [4,11].

In conflict-affected states like South Sudan and CAR that experience protracted humanitarian crises, non-governmental organizations (NGOs) and international agencies have implemented various community engagement programs that included community health workers (CHWs) and need assessments to develop tailored interventions [12]. Communities and community health workers play a key role in this continuum of emergency to long-term response and recovery [13]. Despite global emphasis on localization and community participation in humanitarian settings, there is limited research on how community engagement can support health interventions across different stages of a response cycle in conflict-affected contexts, make these interventions more sustainable, and lead to a more resilient health system.

The Disaster Response Management (DRM) Cycle is an approach used by the various humanitarian organizations. It consists of four interrelated components - preparedness, response, recovery and resilience [14]. Although these components often overlap, it is important to keep these four response aspects in mind while providing health services in fragile and conflict affected settings, as it aids in designing and implementing programs for both acute and prolonged crises while ensuring service continuity and smooth transitions across different phases of the crisis cycle. Despite existing efforts, there are significant knowledge gaps regarding the crucial role of community engagement throughout the DRM cycle to ensure continued health

service delivery in conflict-affected settings while ensuring the safety of the local actors, and how it can influence the humanitarian response beyond service delivery.

This study examines the role of community engagement across the DRM cycle in conflict-affected contexts. Using a qualitative research approach, it explores how community engagement contributes to preparedness, response, and recovery within humanitarian programs. The study also provides recommendations for integrating community-led approaches in humanitarian health interventions to support resilience of health systems.

## Method

### Setting

The International Committee of the Red Cross (ICRC) and the Canadian Red Cross (CRC) in partnership with National Red Cross Societies piloted an integrated public health approach in South Sudan and the Central African Republic (CAR), called the Advanced Partnership in Health (APiH) program (2019–2023), to deliver essential healthcare services to communities affected by ongoing conflict. This initiative aimed to establish a multi-year collaboration between the three partners, local health authorities, and communities to deliver essential evidence-based community-based reproductive, maternal, neonatal, child, and adolescent health (RMNCAH) services in conflict-affected areas while ensuring a continuum of care. CRC research team and implementers from CAR, South Sudan, and Canada were initially engaged to assess the feasibility of the program, particularly the community-based services delivery in these conflict settings using qualitative methods—including key informant interviews and focus group discussions As part of this initiative, a recently published paper [15]highlighted the critical role of CHWs in delivering care in these challenging settings. The study, which was based on the APiH project, also found that there were various challenges to service delivery including security concerns, literacy and language challenges, and limited access to context-specific resources [15]. Building on the primary data collected for that study, this paper examines the role of community engagement across the DRM cycle in conflict settings, particularly its contribution to preparedness, response and recovery.

### Study design

The study adopted a qualitative approach and conducted key informant interviews (KII) and focus group discussions (FGDs) in four districts where the APiH pilot programs were underway: two in CAR (Ouandaogo and Grevai) and two in South Sudan (Ngo Dakala and Ngo KU). This study applied a qualitative descriptive design, guided by framework analysis [16]. This approach allowed for both deductive coding based on DRM phases and inductive coding from participant narratives. The initial study [14] used purposive sampling to select program implementers and CHWs, complemented by maximum variation sampling to capture gender, age, and role diversity.

### Data collection

This study performed a secondary analysis of data collected from a primary study [14] based on the APiH program (2019–2023) to understand how community engagement could facilitate continuity of healthcare services during different phases of crisis. To ensure transparency and clarity, we have briefly described the primary data collection process below.

The primary data were collected from program implementers and community members in South Sudan and CAR using a purposive sampling methodology. Researchers identified program implementers from CAR Red Cross, South Sudan Red Cross, CRC, and ICRC and conducted key informant interviews (KIIs). Experienced field health teams facilitated participant recruitment and conducted focus group discussions (FGDs) with community health workers (CHWs), volunteers, elders, men, women, and adolescents. Open-ended, semi-structured interview guides were used for KIIs and FGDs (available as supplements). The KII and FGD guides covered questions on both community engagement as well as other aspects of APiH project. The specific questions about community engagement were - who the key stakeholders are and

their roles, how they have been engaged, challenges faced and strategies adopted to facilitate engagement. The dataset included 15 KIIs and 16 FGDs involving 169 participants across CAR and South Sudan (Tables 1 and 2). Among the 15 KIIs, two were conducted with women, however, more women participated in the focus group discussion. KIIs were conducted by trained CRC researchers with local Red Cross support, while FGDs took place in accessible community spaces such as health posts or local halls. Sessions lasted 45–60 minutes for KIIs and 60–90 minutes for FGDs. All facilitators had prior training in qualitative research methods, refreshed through CRC-led workshops on interviewing, consent, and psychological first aid. Guides were co-developed by CRC researchers and local health staff, pilot tested for contextual relevance and refined accordingly. Sessions were audio-recorded with consent, supplemented by field notes to capture

**Table 1. Participant characteristics for KII (Rab et al., 2023).**

| Total participants | South Sudan (n = 7) | | CAR (n = 8) | |
|---|---|---|---|---|
| Roles* | Male | Female** | Male | Female** |
| MoH | 2 | 0 | 1 | 0 |
| ICRC | 2 | 0 | 1 | 2 |
| HNC | 1 | 1 | 2 | 0 |
| CRC/ICRC delegate in the field | 1 | 0 | 2 | 0 |

**Table 2. Participant characteristics for FGDs (Rab et al., 2023).**

| 12 FGDs in South Sudan, Total participants: 114 | | | |
|---|---|---|---|
| | No of FGDs | Total participants | Location |
| CHWs/ Volunteers | | | |
| Mixed group (Males and Females) | 2 | 20 (Female:6, Male: 14) | Ngo Dakala (1) Ngo Ku (1) |
| Community leaders/elders | | | |
| Mixed group (Males and Females) | 2 | 20 (Female: 5, Male:15) | Ngo Dakala (1) Ngo Ku (1) |
| Community members | | | |
| Male | 2 | 19 | Ngo Dakala (1) Ngo Ku (1) |
| Female | 2 | 15 | Ngo Dakala (1) Ngo Ku (1) |
| Adolescent boys | 2 | 20 | Ngo Dakala (1) Ngo Ku (1) |
| Adolescent girls | 2 | 20 | Ngo Dakala (1) Ngo Ku (1) |

| 4 FGDs in CAR, Total participants: 40 | | | |
|---|---|---|---|
| | No of FGDs | Total participants | Location |
| CHWs/ Volunteers | | | |
| Male | 1 | 10 | Ouandaogo |
| Female | 1 | 10 | Ouandaogo |
| Community leaders/elders | | | |
| Male | 1 | 10 | Ouandaogo |
| Female | 1 | 10 | Ouandaogo |

*MoH Ministry of Health, ICRC International Committee of the Red Cross, HNS Host National Society, CRC Canadian Red Cross

**Preference was given to female participants, limited in recruitment of female participants for KII as the sampling pool for females in leadership position was limited

non-verbal cues. Data collection took place between in May and June 2021. Transcripts were translated and cross-checked for accuracy.

The KIIs with program implementers were conducted over Zoom and Teams videoconferencing by the CRC research team. The interviews with program implementers in South Sudan were conducted in English, and those in CAR in French. The transcripts in French were transcribed and translated into English by CRC research team members proficient in both languages. The research team included both external researchers and local staff. This insider–outsider combination enhanced cultural sensitivity while ensuring methodological rigor. Local facilitators led interviews, while external researchers supervised and triangulated findings. Regarding the FGDs, experienced health field teams in respective contexts facilitated participant recruitment, conducted FGDs with community members, and managed transcription and translation. In South Sudan, FGDs were conducted in both Ngo Dakala and Ngo Ku; in CAR, FGDs were only conducted in Ouandaogo.

As the data collection was scheduled to happen during the COVID-19 pandemic, the research team at CRC and CRC field staff (delegates) conducted French and English "Training of Trainers" (ToT) including some sessions online with CAR and South Sudan teams, which included training on preparing, conducting and facilitating FGD, including the informed consent and assent process, note-taking, maintaining data security and transcribing audio recordings. The ToT also included a psychological first aid (PFA) component to train the field team how to identify a participant in distress and provide PFA in a timely manner.

## Data analysis

The data were analysed using a framework method of analysing qualitative data [16] as it can be used with various qualitative approaches to generate themes. This method also facilitated both deductive and inductive coding and constant comparative techniques. Using the previously transcribed and translated dataset, the analysis was conducted in the following stages: familiarisation with the data, coding, developing a working analytical framework, applying the analytical framework, charting the data into the framework matrix, and interpreting the data. Constant comparison was applied iteratively, with each new transcript compared against earlier ones to refine codes and ensure consistency across CAR and South Sudan contexts.

Framework analysis allowed for examining the data across the KIIs and FGDs, as well as within the different stages of the DRM cycle. At first, the researchers familiarized themselves with the data by repeatedly reading the transcripts. The deductive codes were identified from the DRM cycle stages which were response, recovery, resilience and preparedness. The inductive coding was undertaken using an open-coding method guided by a constant comparative approach. The data was coded across all applicable stages of DRM cycle and the role of community engagement in health service delivery as it emerged from the data. After coding the first few transcripts, the researchers discussed the deductive and inductive codes and agreed on a coding framework that was applied to the rest of the transcripts. We employed the Framework Method to analyze our qualitative data, following a systematic process that included transcription, familiarization, coding, development of an analytical framework, indexing, charting, and interpretation [16]. This approach facilitated transparent and collaborative analysis across the research team, allowing for structured comparison and thematic exploration aligned with our study objectives.

## Trustworthiness

To ensure the trustworthiness of the study, we applied multiple strategies aligned with qualitative research standards. Credibility was supported through triangulation of data sources (KIIs and FGDs), involvement of both insider and outsider researchers, and iterative discussions during analysis to validate interpretations. Dependability was maintained by keeping a detailed audit trail, including documentation of data collection procedures, transcription and translation processes, and analytic decisions. Confirmability was enhanced through reflexive journaling and team debriefings to minimize researcher bias and ensure that findings were grounded in participants' narratives. Transferability was supported

by providing rich contextual descriptions of the study settings, participant characteristics, and data collection processes, allowing readers to assess the applicability of findings to similar conflict-affected contexts.

## Positionality

The research team comprised both external researchers from the Canadian Red Cross (CRC) and local staff from the South Sudan Red Cross and CAR Red Cross, forming an insider–outsider collaborative model. This positionality was instrumental in ensuring cultural sensitivity, contextual relevance, and ethical rigor throughout the study. Local facilitators, familiar with community dynamics and languages, led the FGDs and supported KIIs, while external researchers provided methodological oversight and triangulated findings. This combination helped mitigate power imbalances, foster trust with participants, and enrich data quality by integrating diverse perspectives.

## Ethics approval

The study received approval (DP_DR 21/00011/ESV/abg) from the ICRC Ethics Review Board. The complete study protocol was shared with the Ministries of Health in South Sudan and CAR and approved by host Red Cross/Red Crescent National Societies in both countries. Written informed consent was obtained from each study participant before initiating each key informant interview. Verbal informed consent was obtained from each participant before initiating the study.(The FGD participants provided verbal consent before the FGD started which was recorded). The study procedures and methods were conducted in accordance with the ethical principals and guidance in the World Medical Association Declaration of Helsinki.

## Findings

From our analysis, we identified various themes that highlight the crucial importance of community engagement in all phases—from pre-disaster preparedness to response, recovery, and resilience building. This engagement is essential for effectively mitigating the impacts of protracted crises while ensuring accountability in service delivery.

Moreover, we explored the overlapping functions of community engagement within conflict settings as part of complex humanitarian programming. We identified a continuum of community engagement that spans all phases of the DRM cycle in conflict-affected areas. We identified five key thematic functions: (1) Building trust between communities and health actors to facilitate access and cooperation; (2) Ensuring community voice and power through inclusive participation in planning, implementation, and evaluation; (3) Promoting community health services by recruiting and supporting community health workers (CHWs) as trusted intermediaries; (4) Enhancing community-led resource mobilization to strengthen local capacity and service continuity; and (5) Fostering resilience through sustained engagement and empowerment.

These functions were found to be interdependent and critical for effective preparedness, response, recovery, and long-term resilience in complex humanitarian programming. The relationship of these five themes to the DRM cycle will be discussed following the description of the findings related to each theme.

### Building trust in a conflict context affected areas to ensure access to health

In conflict-affected regions like South Sudan and the CAR, sustained community engagement is essential for ensuring access to healthcare. The APiH program demonstrated that involving community leaders and members throughout the planning and implementation process fosters trust and ownership. As one South Sudanese elder noted:

*"The elders have a role in understanding the project, understanding what activities this project will implement and where communities stand in those activities, and how communities can benefit on these activities."* (KII S7, South Sudan)

One of the aspects of the APiH project was selecting CHWs from the community to provide information and services. Selecting care provider from the community helped to building relationships with the community by engaging with leaders and elders. This continuum of engagement—especially through respected local figures—helps bridge gaps between health providers and communities to communicate with providers, armed groups and community members to provide clear messaging and description of intervention's goals.

> "The elders have a role in understanding the project, understanding what activities this project will implement and where communities stand in those activities, and how communities can benefit on these activities." (KII S7, South Sudan)

In the focus groups with women in South Sudan, the respondents mentioned communicating and involving the community members from the beginning has contributed to ensuring accountability as well.

> "We tell them we're going to do 123, and then we also listen to them to their suggestion, and they get their feedback related to what has been said." (KII S1, CAR)

Respondents also mentioned that specific and clear messaging about what to do during various crisis moments guided them to seek help and treatment. Moreover, if the project and intervention is clearly explained to the community leaders and elders, and the activities are agreed on prior to commencing the project, it can help to proceed with the intervention. In all settings, but particularly in the conflict settings, it is not possible to proceed if they do not agree.

> "In conducting any activity in the community you have to … first, engage the community. If there is commitment and full engagement of the community leaders and the community key members and the community members in conducting the assessment that would be easier, and it will lead to a better result of the assessment." (KII S3, South Sudan)

### Promoting harmony and resolving conflict in fragile setting

The findings from the study demonstrated that in conflict and fragile settings, actively engaging in dialogue and fostering reconciliation initiatives to build trust between different groups while providing essential services and empowering local leaders, could contribute to promoting harmony and security.

Although security issues and armed groups pose many challenges, the CHWs who were conducting the community engagement activities mentioned that armed groups' co-operation is important to ensure that CHWs feel secure.

> "Health service is a humanitarian work which doesn't need to involve harms to health workers, so armed groups must always enable health workers to take services to the affected communities because health workers are impartial." (FGD with CHW, South Sudan)

In addition, community-based conflict resolution involves community members identifying conflict causes and working towards a solution. it is crucial to engage community members in the discussions as it can help to sensitize community as well as resolve disputes.

> "In case of any dispute, the parties in the dispute can sit together and community members, talk about the issue, resolve the issue, the offender is forgiven and warned not to repeat the same mistake in future." (FGD with men in South Sudan)

## Ensuring inclusion of community voice and power to promote ownership and sustainability of interventions in conflict settings

To ensure inclusivity, it's crucial to engage diverse groups, use accessible communication, encourage open dialogue, and empower individuals to share perspectives, especially in representative selection and decision-making. Village leaders, youth and women engagement is critical to ensure that the community understands the respective roles of the community members, which can promote ownership and contribute to sustainability.

> "For any project to begin in any community you have to meet the community leaders. You engage them from the beginning, from the process of planning to implementation. You involve them in monitoring, at the same time, you have to involve them in the evaluation of the, of the project. So by involving them… you make them own the project as if it is theirs. So this can help in sustainability of any project in the community." (KII S3, South Sudan)

> "... positioning the women's role for the project,.... So we build a kind of like an awareness in the community that having women's as, uh- in an engagement for this project is far more beneficial.... (KII S7, South Sudan)

The adolescent members of the community voiced that they want to be more engaged in the planning and implementation of activities. The village leaders also agreed that community members need to be more aware and empowered to understand the activities that are planned and the goal of those activities. They also felt a sense of responsibility about disseminating the information.

> "The people must be empowered to understand. Once people understand what we are supposed to do, it will be easy for us to achieve." (Village leader, South Sudan)

## Promoting community health services by supporting CHWs

Community members are experts on their own challenges and solutions. Engaging them is key to sustaining health services, especially by recruiting and supporting CHWs.
In conflict-affected areas, it is crucial to work with community members to ensure health access during emergencies -

> "...in areas affected by violence, if any community member is injured or killed, those around can rush to the nearest unaffected community and report the incident, community members can [be] mobilize and rush to the scene of the incident, the wounded person can be carried to safety for treatment, and the dead can be buried." (FGD with men in South Sudan)

CHWs were the actors who associated closely with the community as they conducted the community engagement activities. As such, CHWs emphasized that anticipating potential challenges and planning ahead of time to disseminate information to leaders can contribute to preparedness for health service delivery.

> "...to train people in the community, first I have to send information, the chiefs and community leaders must be informed, once I reach there, I will explain to the community about the program and ask their consent if they are willing to participate in the program or not? If they approve, we can continue, if they say no, we can reschedule. "(FGD with CHW, South Sudan)

Moreover, providing training to CHWs can help them feel empowered and work towards the improvement of healthcare services and community engagement activities

"...Training empowered us to support in responses to emergency (FGD, CHW, South Sudan)

The community is also advocating for CHWs and mentioned that the CHWs are working to improve the services, but their incentive should be increased, their security issues should be considered, and they should be better supported.

"...the staff are delivering to our satisfaction and serving with courtesy but we feel there is a need to increase their salaries to motivate them further." (FGD with men in South Sudan)

One of the community leaders mentioned,

"During outbreaks of violence, there should be immediate emergency response planning by the community and the health personnel on how to move personnel together with the community and medicines to safety" (Community leader at Ngudakala/ South Sudan)

**Enhancing community-led resource mobilization: For capacity strengthening and service provision**

Empowering communities through capacity strengthening and logistical support can improve local involvement in service delivery. Community based organization including the local branches of Red Cross and local partners in the conflict area are vital in reaching remote areas, offering services, and supporting infrastructure development. Logistics are crucial to ensure CHWs can access hard-to-reach areas, with advanced planning for personnel and equipment movement during emergencies. They also mentioned that storage facilities should be built so that drugs and supplies can be stored for longer period and can be transported to other facilities.

".... Red Cross should plan to build storage facility which can be able to store drugs for at least six months. Supplies from such a store can be transported to other localities." (FGD, Village leader, South Sudan)

CHWs also mentioned that expanding the role of CHWs in the conflict setting - for example, with additional training on topics like mental health could help deliver services to people who are suffering from various mental health issues but do not have the access to services due to conflict and insecurity.

"We need mental health and counseling program here and more volunteers to be trained to do outreach because people are suffering from trauma and don't know what to do." (CHW from South Sudan)

Participants mentioned that in some cases, only the Red Cross is able to reach remote or insecure areas to offer services. They also mentioned that training local community members and improving availability of essential medicine and medical supplies can help them to provide better services.

"There's a need to train the locals with skills on basic health services such as injections so that when the Red Cross team are not close, the locals can provide the basic needed services." (FGD with men in South Sudan)

"If there is a possibility, let the Red Cross take services to communities in distanced locations who are unable to access services here because of insecurity." (Community leaders in South Sudan)

Community-based approaches and strategies with the organisational tools and resources can carry out such changes in conflict and fragile settings.

**Engaging community throughout DRM cycle to ensure continuity of health care in conflict affected context**

Our study underscores the critical and overlapping role of community engagement in conflict zones, highlighting its overlapping impact across the DRM cycle. The findings reveal that community engagement plays a pivotal role in pre-disaster preparedness, response, recovery, and community resilience building.

Community engagement is crucial in building trust among conflicting parties by facilitating open communication and acknowledging diverse perspectives. This trust is essential for effective disaster preparedness, as it ensures cooperation during crises. In the recovery phase, this trust supports cooperation and aids in rebuilding efforts. Ultimately, long-term trust enhances community resilience, enabling better management of future conflicts and disasters.

Engaging communities in dialogue and reconciliation initiatives during crises helps resolve disputes and promotes peace, ensuring better disaster preparedness and coordinated responses. This engagement fosters a peaceful environment that supports recovery efforts. Continuous peace-building activities further enhance resilience, preparing communities for future conflicts and disasters.

Engaging diverse groups in planning and decision-making ensures inclusivity and empowers individuals, which is crucial for disaster preparedness and response. Empowering community members in recovery efforts addresses their needs and concerns, leading to sustainable rebuilding. Continuous involvement in decision-making processes strengthens ownership and responsibility, enhancing community resilience.

The community engagement activities are conducted by CHWs who have a crucial role to play as an intermediary between the intervention actors and the community. Recruiting CHWs from the community and supporting them is essential for sustaining health services and emergency preparedness. Training and supporting CHWs during recovery empower them to improve healthcare services and enhances community resilience by ensuring sustained healthcare services and preparedness for future emergencies.

Empowering communities through capacity strengthening of CHWs and providing logistical support strengthens local involvement in service delivery and disaster preparedness. Training local community members in basic healthcare services such as first-aid and improving the availability of essential medicine and supplies enhance resilience and preparedness for future disasters.

The findings highlight that community engagement is crucial across all phases of the DRM in conflict zones. It builds trust, promotes peace, ensures community voice, supports health workers, and enhances resource mobilization. These efforts are essential for effective disaster preparedness, response, recovery, and resilience building, ultimately fostering sustainable and resilient communities.

## Discussion

In this paper, we aimed to highlight the role community engagement to demonstrate that engaging communities throughout the DRM cycle is crucial for ensuring the continuity of healthcare in conflict-affected contexts. In light of the recent changes in the humanitarian aid landscape, there is increased attention on localization of services in humanitarian crises for transitioning to long-term recovery from emergency response [17,18]. Evidence also suggest that community engagement can lead to responsive and community-driven health interventions in a crisis setting [19]. The experiences of the Red Cross in South Sudan and the CAR provide valuable insights into effective community engagement strategies that can guide humanitarian health responses in similar settings.

With attacks on healthcare increasing exponentially, the WHO is documenting 1,479 incidents across 32 conflict-affected countries in 2022 [20]. Evidence suggests that community engagement—through dialogue with local actors, transparent communication, and building local acceptance—can mitigate the risks of disrupting health services and eroding trust, and enhance the safety of both health workers and affected populations [9,21]. Recent studies emphasize the importance of integrating community engagement in humanitarian action to address the needs and initiatives of affected communities, ensuring that interventions are inclusive and context-specific [4,22]. Studies highlight that when communities

are engaged in response and recovery phases of humanitarian programming, it can improve health service utilization and increase access for diverse communities by building trust and integrating local knowledge into program design and delivery [23,24]. The Red Cross's community health programs in South Sudan and CAR have demonstrated the feasibility of delivering healthcare services in conflict settings through a community-based approach. These programs highlight the significance of clear messaging, community inclusiveness, and localized plans for service delivery [25,26].

Previous research has stated that community engagement provides opportunity for the community to share their perspectives and guide how health services should be offered in a culturally sensitive way and develop ownership from the community [11,27]. In South Sudan, the Red Cross has successfully engaged local communities to build trust and cooperation. Training CHWs and volunteers has empowered them to disseminate health information and provide first aid. This proactive engagement ensures that communities are better prepared to handle emergencies [26]. Additionally, negotiating safe passage for healthcare delivery in conflict zones has underscored the importance of collaboration and trust-building with local actors [25].

In CAR, community engagement has been pivotal in promoting peace and conflict resolution. The Red Cross's efforts to involve community members in discussions and decision-making processes have helped address immediate health needs and foster a sense of ownership and responsibility [25,26]. Continuous support for CHWs and the provision of context-specific resources have mitigated barriers to service delivery, improving healthcare access and strengthening community resilience [25].

Previous research has stated that community engagement can create deliberative spaces for community involvement that can ensure feedback mechanisms and equal representation of all groups [27,28]. Beyond South Sudan and CAR, other case studies further illustrate the impact of community engagement in humanitarian settings. In Nigeria, the Red Cross's community engagement efforts during the Lassa Fever awareness campaign demonstrated the effectiveness of involving local communities in public health initiatives. By setting up community committees for feedback and complaints, the campaign ensured that community members were actively involved in the response, leading to increased awareness and better health outcomes [29].

In Mali, the Malian Red Cross integrated community engagement into all its activities through workshops and collaborative efforts. This approach helped build trust and cooperation among community members, enhancing the effectiveness of humanitarian interventions (Community Engagement Hub, 2023). Similarly, in Eswatini, the Baphalali Red Cross Society's Early Action Protocol validation exercise involved community feedback to plan anticipatory actions for drought-affected areas. This engagement ensured that the community's needs and perspectives were considered, leading to more effective and timely interventions [30].

In other regions, such as Syria and Yemen, community engagement has been crucial in supporting intervention in the complex emergencies arising from prolonged conflicts and natural disasters. Integrating social sciences into community engagement has been recognized as valuable for inclusive humanitarian programming, aiming to be attentive to the needs and initiatives of affected communities [22,31]. Community engagement helps tailor services to the specific context, recognizing that community priorities may differ from those of external experts, particularly those unfamiliar with local realities [11,27]. However, challenges such as security risks, operational difficulties, and funding constraints persist, hindered effective engagement [4,9,22].

Despite these challenges, there are significant opportunities for enhancing community engagement. Leveraging local knowledge and networks can improve the effectiveness of interventions. Collaborating with local authorities and organizations can foster trust and cooperation, which may improve healthcare outcomes in crisis-affected areas, enhancing the sustainability and relevance of interventions [11,32]. Utilizing technology for communication and data collection can enhance the reach and impact of community engagement efforts [4,31,33,34].

To ensure effective community engagement in humanitarian settings, several strategies can be employed: engaging diverse community groups in planning and decision-making processes to ensure inclusivity; providing training and support

to community health workers and volunteers to empower them and improve service delivery; using clear and specific messaging to guide community members during crises; collaborating with local authorities, organizations, and community leaders to build trust; and leveraging technology for communication, data collection, and monitoring [9,22,25].

Effective community engagement has several policy and practice implications. Policies should prioritize community engagement as a core component of humanitarian response, ensuring that interventions are inclusive and context specific. Adequate resources should be allocated to support community engagement initiatives, including funding for training, capacity building, and technological tools. Implementing robust monitoring and evaluation frameworks can help assess the impact of community engagement and identify areas for improvement [4,22]. Promoting sustainable practices that empower communities and build resilience can ensure long-term benefits and preparedness for future conflicts and disasters [9,31].

In conclusion, the experiences of the Red Cross in South Sudan and CAR, along with other case studies from Nigeria, Mali, Eswatini, Syria, and Yemen, highlight the critical role of community engagement throughout the DRM cycle. By building trust, promoting peace, ensuring community voice, supporting health workers, and enhancing resource mobilization, humanitarian organizations can ensure the continuity of healthcare in conflict-affected contexts. These strategies provide a valuable framework for guiding humanitarian health responses in similar settings.

**Policy Recommendations**

1. **Institutionalize community engagement** as a core strategy across all DRM phases in humanitarian health programming.

2. **Develop conflict-sensitive engagement frameworks** that reflect local dynamics, cultural norms, and power structures.

3. **Strengthen CHW systems** by recruiting from within communities, providing training in emergency response and ensuring safety and fair compensation.

4. **Ensure inclusive participation** by engaging women, youth, and marginalized groups in decision-making and monitoring processes.

5. **Establish community feedback and accountability mechanisms** to improve transparency, responsiveness, and trust.

6. **Invest in local infrastructure and logistics**, including storage and transport systems, to support community-led service delivery in hard-to-reach areas.

Integrating these strategies will not only improve the effectiveness and equity of humanitarian health responses but also build more resilient, community-driven health systems in fragile and conflict-affected contexts.

## Conclusion

This study demonstrates that community engagement is a critical enabler of health service continuity across all phases of the Disaster Risk Management (DRM) cycle—preparedness, response, recovery, and resilience—in conflict-affected settings. Drawing on evidence from South Sudan and the Central African Republic, the findings highlight five interrelated functions of community engagement: building trust, amplifying community voice, supporting community health workers (CHWs), mobilizing local resources, and fostering resilience.

Community health workers' engagement in DRM enhances preparedness by involving local actors, including the community in planning and training CHWs. During response, they facilitated access to services through trusted intermediaries. In recovery, inclusive participation promoted ownership and sustainability. For resilience, community-led initiatives strengthened local capacity and adaptability.

## Acknowledgments

The authors extend their sincere gratitude to all key informants and focus group participants who generously shared their time, perspectives, and valuable insights, which were instrumental to this study. We are also deeply appreciative of all study participants for their thoughtful contributions. Special thanks go to the Canadian Red Cross team, the South Sudan and Central African Republic National Society teams for their vital contributions to the development of this project, and coordination of the data collection, as well as to our colleagues at ICRC-Geneva for their support.

## Author contributions

**Conceptualization:** Akalewold T. Gebremeskel, Puspita Hossain, Salim Sohani.

**Data curation:** Akalewold T. Gebremeskel, Puspita Hossain, Ilja Ormel, Faiza Rab, Mekdes Assefa, Christina Angelakis, Mariam Kone, Salim Sohani.

**Formal analysis:** Akalewold T. Gebremeskel, Puspita Hossain, Ilja Ormel, Mekdes Assefa, Christina Angelakis, Mariam Kone, Salim Sohani.

**Funding acquisition:** Akalewold T. Gebremeskel, Ilja Ormel, Faiza Rab, Mariam Kone, Salim Sohani.

**Investigation:** Akalewold T. Gebremeskel, Puspita Hossain, Ilja Ormel, Faiza Rab, Mariam Kone, Salim Sohani.

**Methodology:** Akalewold T. Gebremeskel, Puspita Hossain, Ilja Ormel, Faiza Rab, Mekdes Assefa, Christina Angelakis, Mariam Kone, Salim Sohani.

**Project administration:** Akalewold T. Gebremeskel, Ilja Ormel, Faiza Rab, Mariam Kone, Salim Sohani.

**Resources:** Akalewold T. Gebremeskel, Ilja Ormel, Faiza Rab, Mekdes Assefa, Christina Angelakis, Mariam Kone, Salim Sohani.

**Software:** Akalewold T. Gebremeskel, Puspita Hossain.

**Supervision:** Akalewold T. Gebremeskel, Salim Sohani.

**Validation:** Akalewold T. Gebremeskel, Puspita Hossain, Ilja Ormel, Faiza Rab, Mekdes Assefa, Christina Angelakis, Mariam Kone, Salim Sohani.

**Visualization:** Akalewold T. Gebremeskel, Puspita Hossain.

**Writing – original draft:** Akalewold T. Gebremeskel, Puspita Hossain.

**Writing – review & editing:** Akalewold T. Gebremeskel, Puspita Hossain, Ilja Ormel, Faiza Rab, Mekdes Assefa, Christina Angelakis, Mariam Kone, Salim Sohani.

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
