## [Decision Letter · Decision Letter 0]

1 Oct 2025

PGPH-D-25-01874

Continuum of community engagement to ensure access to health care in conflict-affected areas in South Sudan and Central African Republic (CAR): Lessons from the Red Cross

Dear Dr. Gebremeskel,

Thank you for submitting your manuscript to PLOS Global Public Health. After careful consideration, we feel that it has merit but does not fully meet PLOS Global Public Health’s publication criteria as it currently stands. Therefore, we invite you to submit a revised version of the manuscript that addresses the points raised during the review process.

I would like to sincerely apologise for the delay you have incurred with your submission. It has been exceptionally difficult to secure reviewers to evaluate your study. We have now received two completed reviews; the comments are available below. The reviewers have raised significant scientific concerns about the study that need to be addressed in a revision.

Please revise the manuscript to address all the reviewer's comments in a point-by-point response in order to ensure it is meeting the journal's publication criteria. Please note that the revised manuscript will need to undergo further review, we thus cannot at this point anticipate the outcome of the evaluation process.

We look forward to receiving your revised manuscript.

Kind regards,

Miquel Vall-llosera Camps

Staff Editor

Journal Requirements:

Reviewers' comments:

Reviewer's Responses to Questions

**Comments to the Author**

1. Does this manuscript meet PLOS Global Public Health’s publication criteria?

Reviewer #1: Yes

Reviewer #2: Yes

2. Has the statistical analysis been performed appropriately and rigorously?

Reviewer #1: N/A

Reviewer #2: N/A

3. Have the authors made all data underlying the findings in their manuscript fully available (please refer to the Data Availability Statement at the start of the manuscript PDF file)?

Reviewer #1: Yes

Reviewer #2: No

4. Is the manuscript presented in an intelligible fashion and written in standard English?

Reviewer #1: Yes

Reviewer #2: Yes

Reviewer #1: Thank you for this excellent work and paper. As humanitarian crises increase around the world these findings are crucial. As a previous humanitarian worker, I appreciated this paper and the potential policy impact it can have.

My suggestions are to improve the flow, situate the paper and its findings in contemporary context, and to improve clarity in the methods section.

Background:

1. Include some statistics in the introduction. How much has it increased? See the data here: https://www.prio.org/news/3616#:~:text=from%20world%20stage-,New%20data%20shows%20conflict%20at%20historic%20high%20as%20U.S.%20signals,the%20Uppsala%20Conflict%20Data%20Program

Conflicts at an all time high since World War II

Methods:

2. Minor: Use KII after initial abbreviation

3. In the methods section, a table summarizing the interviewee profile data would be helpful. You can indicate the regions, the types of KI categories (e.g., CHWs, community elders, community adolescents etc. and include the number of KIs per each category). Similarly, for FGDs, number of participants, the composition of the group (a mix of what kind of participants based on the types you have described in the paper).

4. You have indicated you analyzed previous data – tell us a bit more. Maybe it should come before you describe your original data collection? Or do you mean for this study, you analyzed previously collected data? I think that detail should perhaps come at the top. This study either analyzed previous data to set a foundation for our KIIs and FGDs and then describe your study design OR This study was based on previously collected KII and FGD data.

5. Please add how many KIIs and FGDs were conducted in total maybe at the end of data collection section or somewhere relevant in the data collection section.

6. I would consider moving Tables 1 and 2 from findings to study design section because it is less findings more study participant detail. This can address the comments number 3 and 5.

7. My suggestion would be to move the tables to the study design section, then combine the analysis sections (data analysis and analysis of the collected data sections together). Then, have the results or findings section after. Might be a better flow.

8. Minor suggestion: It is good to say, we identified five key thematic functions than just saying they emerged. This is stylistic but helps to connect clearly with what you planned under the framework method.

Findings of qualitative interviews:

9. I think the policy recommendations should come at the end of the results section. Definitely not at the conclusion. You can even point to the recommendations in the discussion.

Discussion

10. I think it is crucial to have a paragraph on rising attacks on healthcare and civilians in the discussion. Discuss how community engagement can help mitigate trauma and build trust in an increasingly hostile and dangerous environment.

Reviewer #2: The strenght of the article are

- Provides comprehensive coverage across the DRM phases.

- Presents clear and meaningful themes.

- Qualitative data are thick and rich

- Engages with a wide evidence base in the discussion.

- Highlights policy and practice implications effectively.

- Shows strong integration with existing literature.

The comments are

1.The study does not clearly state which qualitative approach guided the research design—whether it was phenomenology, grounded theory, or ethnography.

2.Greater detail is needed on the conduct of KIIs and FGDs:

• Who conducted them?

• Where were they held?

• Were the interviewers trained in qualitative methods?

• What was the typical duration of each session?

3. The process of developing the interview guide is not described.

4.Information about the researchers themselves is missing—whether they were insiders from the study region, and how this positionality might have influenced data collection.

5.The use of constant comparative techniques is mentioned, but requires elaboration on how exactly this was applied.

6. Although the authors state that framework analysis was used, it is unclear whether the framework was developed or adapted. Further explanation is needed on the analytic pathway—e.g., how data were coded, charted, or mapped against the DRM cycle—to demonstrate methodological rigor.

7.The strategies for ensuring trustworthiness (credibility, dependability, confirmability, transferability) are not sufficiently discussed.

8.Some sections (e.g., quotes from South Sudanese elders) appear to be repeated verbatim and should be streamlined.

Conceptual foundations of the study

9. Approaches used for sampling

10. Techniques applied to document the data (e.g., audio recordings, video recordings, or written field notes)

11. Proportion of individuals who did not participate

**Do you want your identity to be public for this peer review?** For information about this choice, including consent withdrawal, please see our Privacy Policy

Reviewer #1: No

Reviewer #2: No

---

## [Decision Letter · Decision Letter 1]

21 Dec 2025

Continuum of community engagement to ensure access to health care in conflict-affected areas in South Sudan and Central African Republic (CAR): Lessons from the Red Cross

PGPH-D-25-01874R1

Dear Dr Gebremeskel,

We are pleased to inform you that your manuscript 'Continuum of community engagement to ensure access to health care in conflict-affected areas in South Sudan and Central African Republic (CAR): Lessons from the Red Cross' has been provisionally accepted for publication in PLOS Global Public Health.

Best regards,

Julia Robinson

Executive Editor

Reviewer Comments (if any, and for reference):

Reviewer's Responses to Questions

**Comments to the Author**

Reviewer #2: All comments have been addressed

publication criteria?

Reviewer #2: Yes

3. Has the statistical analysis been performed appropriately and rigorously?

Reviewer #2: N/A

4. Have the authors made all data underlying the findings in their manuscript fully available (please refer to the Data Availability Statement at the start of the manuscript PDF file)?

Reviewer #2: Yes

5. Is the manuscript presented in an intelligible fashion and written in standard English?

Reviewer #2: Yes

Reviewer #2: (No Response)

**Do you want your identity to be public for this peer review?** For information about this choice, including consent withdrawal, please see our Privacy Policy

Reviewer #2: **Yes:** Kalaiselvan Ganapathy
